# Transcriptome Analysis of Nitrogen-Deficiency-Responsive Genes in Two Potato Cultivars

**Qiaorong Wei** [1,2,3,†], **Yanbin Yin** [1,2,4,†], **Bin Deng** [1,†], **Xuewei Song** [1,2], **Zhenping Gong** [1,*] **and Ying Shi** [1,2,*]

1   College of Agriculture, Northeast Agricultural University, Harbin 150038, China
2   Key Laboratory of Germplasm Enhancement, Physiology and Ecology of Food Crops in Cold Region, Ministry of Education, Harbin 150038, China
3   National Key Laboratory of Smart Farm Technology and System, Key Laboratory of Soybean Biology in Chinese Education Ministry, Northeast Agricultural University, Harbin 150038, China
4   Rice Research Institute, Shenyang Agricultural University, Shenyang 110065, China
*   Correspondence: gzpyx2004@163.com (Z.G.); potato@neau.edu.cn (Y.S.)
†   These authors contributed equally to this work.

**Abstract:** The potato is the third largest food crop, and nitrogen fertilizer is important for increasing potato yields; however, the shallow root system of potatoes causes the nitrogen fertilizer utilization rate to be low, which results in waste and environmental pollution, meaning that high nitrogen efficiency breeding is highly significant for potatoes. In the high nitrogen efficiency breeding of potatoes, genes with a nitrogen-deficient response should first be identified, and RNA-seq is an efficient method for identifying nitrogen-deficiency-response genes. In this study, two potato cultivars, Dongnong 322 (DN322) and Dongnong 314 (DN314), were utilized, and two nitrogen fertilizer application rates (N0 and N1) were set for both cultivars. Through the determination of physiological indicators, we identified that DN314 is more sensitive to nitrogen fertilizer, while DN322 is relatively insensitive to nitrogen fertilizer. Samples were taken at the seedling and tuber formation stage. At the seedling stage, DN322 and DN314 had 573 and 150 differentially expressed genes (DEGs), while at the tuber formation stage, they had 59 and 1905 DEGs, respectively. A total of three genes related to a low-nitrogen response were obtained via the combined analysis of differentially expressed genes (DEGs) and weighted correlation network analysis (WGCNA), of which two genes were obtained at the tuber formation stage and one gene in the seedling stage, providing theoretical guidance for the high nitrogen efficiency breeding of potatoes.

**Keywords:** potatoes; RNA-seq; DEGs; WGCNA

## 1. Introduction

The potato (*Solanum tuberosum* L.) is the third largest food crop [1], and is rich in starch, protein, vitamins, and other nutrients [2]. In China, potatoes are both vegetables and staples, so increasing the potato yield is particularly important to meet individual nutritional needs. The use of nitrogen fertilizers is important for increasing potato production [3], but due to the shallow root system of potatoes, the nitrogen fertilizer utilization rate of potatoes is low, generally being less than 50% [4,5], which causes significant waste and produces environmental pollution. The environmental pollution of nitrogen fertilizer is mainly divided into the following aspects: first, the fertilizer pollutes water resources, resulting in water eutrophication; second, it pollutes the soil, causing soil compaction; and third, it increases $N_2O$ emissions, polluting the air [6–9]. In the potato production process, reducing nitrogen input and improving nitrogen fertilizer efficiency are important research topics. It has been proven that a reasonable combination of water and fertilizer [10], or the use of slow-release nitrogen fertilizer, can improve nitrogen fertilizer utilization [11,12]. However, these two methods require producers to have a greater level of experience, and the cost of

slow-release fertilizer is relatively high; therefore, high nitrogen efficiency breeding seems to be a better way to improve potatoes' nitrogen fertilizer utilization.

In rice and maize, genes associated with nitrogen uptake, utilization, and NUE have been cloned that can increase crop yields or improve crop quality. The nitrate transporter genes NRT1.1b [13], NRT2.1 [14,15], NRT2.3a, and NRT2.3b [16,17] were reported as being able to transfer $NO_3^-$ ions, while the nitrate reductase OsNR2 was reported as being able to reduce $NO_3^-$ ions; these genes increase rice yield by improving NUE. Some transcriptional regulators have also been reported to increase rice yield by improving NUE; the transcriptional regulator Ghd7 directly binds to the promoter of *ARE1* to improve NUE and increase rice yields [18], and OsDREB1C directly binds to the promoters of *NR2*, *OsNRT2.4*, and *OsNRT1.1B* [19] and activates their expression. The overexpression lines of *OsDREB1C* increased yields by 41.3–68.3% [19]. In maize, THP9 increases the corn protein content by improving NUE [20].

However, in potatoes, fewer genes have been reported to affect NUE; In this study, the RNA-seq method was used to identify low-nitrogen-response genes at seedling and tuber formation stages in potatoes. First, a weighted correlation network analysis (WGCNA) was used to create a co-expression module; second, the phenotype and each module were analyzed to determine the nitrogen-deficiency-response module; and third, we conducted an analysis of nitrogen-deficiency-response genes by combining the nitrogen-deficiency-response expression modules with the differentially expressed genes (DEGs). A total of three nitrogen-deficient genes were identified, including one gene at the seedling stage and two genes at the tuber formation stage, which provided theoretical knowledge for the high nitrogen efficiency breeding of potatoes. In this study, we used WGCNA and DEGs to identify three genes related to the nitrogen-deficient response; these are new genes which can be used for potato breeding to improve potato nitrogen use efficiency and enrich the potato nitrogen-response gene pathways.

## 2. Materials and Methods

### 2.1. Plant Materials and Experimental Treatment

Two potato cultivars, Dongnong322 (DN322, with a growth period of 90 days) and Dongnong314 (DN314, with a growth period of 90 days), were planted in pots with soil at the Potato Research Institute of Northeast Agricultural University in Harbin, China. The size of the pots was 33 cm × 60 cm; the volume was 0.51 $m^3$. Detoxified seed potatoes (of about 50 g) of DN322 and DN314 were provided by the Potato Research Institute of Northeast Agricultural University. The two potato cultivars were divided into the N0 (nitrogen deficient) and N1 (normal nitrogen) groups for nitrogen treatment. Plants in the N0 group were treated with 0 g N (urea, N: 46%), 6 g P ($P_2O_5$: 44%), and 6 g K ($K_2O$ 50%); plants in the N1 group were treated with 4.5 g N (urea, N: 46%), 6 g P ($P_2O_5$: 44%), and 6 g K ($K_2O$ 50%). At the seedling and tuber formation stages, leaves were sampled and frozen in liquid nitrogen and stored at −80 °C for RNA-seq.

### 2.2. The Measurement of Plant Dry Matter Weight, Nitrogen Content, and Other Physiological Parameters

At the seedling and tuber formation stages, samples were taken from three plants from both DN322 and DN314 for every N treatment, following which the plants were divided into the root, stem, leaf, and tuber. Methods for determining plant dry matter were based on previous reports [2]; in brief, the divided parts of the potato were heated at 105 °C for 30 min in an oven, and then dried to a constant weight at 70 °C. Total nitrogen content was measured using previously defined methods [21]; in brief, the dry matter of each tissue was ground into powder and passed through an 800-mesh sieve. A 0.2 g sample was taken, and the nitrogen content was measured using a concentrated sulfuric acid–hydrogen peroxide solution, where the total nitrogen content = nitrogen content of different tissue × total dry matter of different tissue. N utilization efficiency (NUE) = ($N1_{NC}$ − $N0_{NC}$)/TNA. $N1_{NC}$ represents the nitrogen accumulation in nitrogen-treated plants, $N0_{NC}$ represents the nitrogen accumulation

in non-nitrogen-treated plants, and TNA represents the total nitrogen application. The net photosynthetic rate was measured using a method described previously [22]; in brief, on sunny days from 9:00 a.m. to 11:30 a.m., the net photosynthetic rate was determined with a Yaxin photosynthesis analyzer (XY-1101). Root vitality was also measured using previously defined methods [23]: First, the standard curve was developed using different concentrations of 1,3,5-triphenylformazan (TTF), and then 0.4% 2,3,5-triphenyl tetrazolium chloride (TTC) and a phosphoric acid buffer were configured into a mixture, where the roots were immersed in the mixture and kept warm for 1–3 h at 37 °C in the dark. Then, 1M sulfuric acid was added to react for 10 min, and finally, the roots were ground with ethyl acetate and a small amount of quartz sand. Root viability was determined according to the standard curve, where root vitality = TTC reduction amount/(root weight × kept warm time).

### 2.3. RNA Extraction, RNA Sequencing, and RT-qPCR Analysis

RNA extraction and RT-qPCR analysis were analyzed via methods described previously [24,25]. In brief, an RNA extraction kit (Promega, Cat.LS1040) was used to extract the RNA according to the manufacturer's method, and 3 μg was used to synthesize the first-strand cDNA (Thermo Fisher Scientific, Cat. K1621). Controlling RNA quantity and quality was performed in the same manner as in a previous study [26]; in brief, the quantity of RNA was measured with the Nanorop2000, whereas the RNA quality was verified using a 1% agarose gel, and after 150 V electrophoresis for 5 min, two sharp bands formed on the gel, indicating that the RNA quality was good. The RNA sequencing method performed was the same as that employed in another previous study [26]; in brief, the RNA sequencing process used was library establishment–library quality control sequencing. This study used the Illumina platform for sequencing, where the number of sequencing bases was Q30 > 91%, and the sequencing quality was ideal. For RT-qPCR, TB green premix Ex Taq (Takara, Cat. RR420A) and a QuantStudio 5 real-time PCR instrument (Applied Biosystems) were used according to the manufacturers' protocols, and *EIF-3e* was used as the internal control gene [27]. In RT-qPCR analysis, we used three biological replicates for different treatments.

### 2.4. Gene Expression Levels and Differentially Expressed Gene (DEG) Analysis

Gene expression levels were estimated using fragments per kilobase of transcripts per million fragments mapped (FPKM), as previously described [28,29]. DEGs between N0 and N1 in DN322 or DN314 were determined with the DESeq2 [30] using a model based on the negative binomial distribution. The resulting *p*-values were adjusted using Benjamini and Hochberg's approach for controlling the false discovery rate (FDR). An adjusted FDR ≤ 0.01 and log2 fold changes ≥1.5 were used as the thresholds of differential expression.

Gene ontology (GO) enrichment analysis of the low-nitrogen-response module genes was implemented via the GOseq R package based on the Wallenius non-central hypergeometric distribution [31], and KOBAS software [32] was used to analyze the low-nitrogen-response module genes for KEGG enrichment [33].

### 2.5. Weighted Gene Co-Expression Network Analysis (WGCNA)

A total of 20,832 and 21,061 genes with an average FPKM > 1 from the seedling and tuber formation stages were selected for the WGCNA network analysis. The WGCNA (v1.29) package in R was used, as in previous studies [34,35]. The module similarity threshold was 0.25, and the minimum number of genes for the module was 30.

## 3. Results

### 3.1. Nitrogen-Deficiency Stress Affects Potato Growth and Development

We first investigated the effect of nitrogen-deficiency stress on dry matter accumulation in potatoes. The dry matter accumulation of DN322 and DN314 at the seedling stage decreased by 60.7% and 78.6%, and that at the tuber formation stage decreased by 45.1% and 80.8%, respectively (Figure 1B,C; Supplementary Table S1). Among them, during the potato tuber formation stage, the dry matter weight of DN322 and DN314 decreased by 45.1% and

80.8%, respectively (Figure 1D), and the dry matter weight of the roots, stems, and leaves of DN322 and DN314 also decreased significantly (Supplementary Figure S1A–F). In addition, nitrogen-deficiency stress also reduced nitrogen accumulation in the roots, stems, leaves, and tubers of DN322 and DN314 (Supplementary Figure S1G–L). We also investigated the effects of low-nitrogen stress on the physiological indicators of DN322 and DN314. At the seedling stage, the root viability of DN322 and DN314 decreased by 15.4% and 41.8%, and the tuber formation stage decreased by 24.0% and 42.5%, respectively (Figure 1E,F; Supplementary Table S1). Photosynthesis is of great significance for potato yield formation, and we investigated the effect of low-nitrogen stress on the net photosynthetic rate of DN322 and DN314, which decreased by 40.1% and 39.7% at the seedling stage and by 41.0% and 40.7% at the tuber formation stage, respectively (Figure 1G,H; Supplementary Table S1). Low-nitrogen stress also significantly reduced the plant height for DN322 and DN314 (Figure 1I).We measured the N utilization efficiency (NUE) of DN322 and DN314, demonstrating that DN314 had a higher NUE (Supplementary Figure S2H). In conclusion, nitrogen-deficiency stress affected potato growth and development, and DN314 is more sensitive to nitrogen than DN322.

### 3.2. High-Throughput RNA-seq to Analyze the Effect of Nitrogen-Deficiency Stress on DN322

To further investigate the effect of nitrogen-deficiency stress on potato molecular mechanisms, high-throughput RNA-seq was performed to analyze the effect of nitrogen-deficiency stress on DN322 at the seedling and tuber formation stages. The clean reads ranged from 20,457,525 to 31,881,882, and the Q30 ranged from 91.25% to 94.32%; the sequencing quality was determined to be ideal (Supplementary Table S2), enabling further analysis. The leaves of DN322 had a total of 21,431 gene expressions at the seedling stage and 573 differentially expressed genes (DEGs) under nitrogen-deficiency stress, including 298 upregulated expression genes and 275 downregulated expression genes (Figure 2A, Supplementary Table S4). We also randomly selected six DEGs and performed RT-qPCR to verify the accuracy of RNA-seq; we found that the expression trend of RT-qPCR was the same as that of RNA-seq under nitrogen-deficiency stress (Supplementary Figure S2A–F). The identified DEGs were annotated with 46 GO terms: 17 biological processes, 17 cellular components, and 12 molecular functions (Figure 2B). Similarly, KEGG annotation was performed, and the identified DEGs were mainly enriched via steroid biosynthesis, diterpenoid biosynthesis, flavonoid biosynthesis, and the valine, leucine, and isoleucine degradation pathways (Figure 2C). During the tuber formation stage, there were a total of 19,905 gene expressions under nitrogen-deficiency stress and 51 differentially expressed genes, of which 15 genes were upregulated and 36 were downregulated (Figure 2D, Supplementary Table S5). The DEGs were annotated with 23 GO terms: 11 biological processes, 7 cellular components, and 5 molecular functions (Figure 2E). The DEGs were also annotated with KEGG, and mainly photosynthesis and oxidative phosphorylation were found to be enriched (Figure 2F).

### 3.3. High-Throughput RNA-seq to Analyze the Effect of Nitrogen-Deficiency Stress on DN314

We also performed RNA-seq on leaves at the seedling and tuber formation stages of the potato cultivar DN314. The clean reads ranged from 19,127,622 to 22,712,674, and the Q30 ranged from 91.38% to 93.88% (Supplementary Table S3). The leaves of DN314 had a total of 20,832 expressed genes at the seedling stage and a total of 150 differentially expressed genes (DEGs) under nitrogen-deficiency stress, including 56 upregulated expression genes and 94 downregulated expression genes (Figure 3A, Supplementary Table S6). The identified DEGs were annotated with 34 GO terms: 13 biological processes, 12 cellular components, and 9 molecular functions (Figure 3B). Similarly, KEGG annotation was performed, and the identified DEGs were mainly enriched via protein processing in the endoplasmic, citrate cycle, and photosynthesis-antenna protein pathways (Figure 3C). During the tuber formation stage, there were a total of 21,061 gene expressions under nitrogen-deficiency stress and 1905 differentially expressed genes, of which 750 genes were upregulated and 1155 were downregulated (Figure 3D, Supplementary Table S7). The DEGs were annotated

with 43 GO terms: 17 biological processes, 15 cellular components, and 11 molecular functions (Figure 3E). The DEGs were also annotated with KEGG regarding photosynthesis, photosynthesis-antenna proteins, carbon fixation in photosynthetic organisms, pentose and glucuronate interconversions, and starch and sucrose metabolism (Figure 3F).

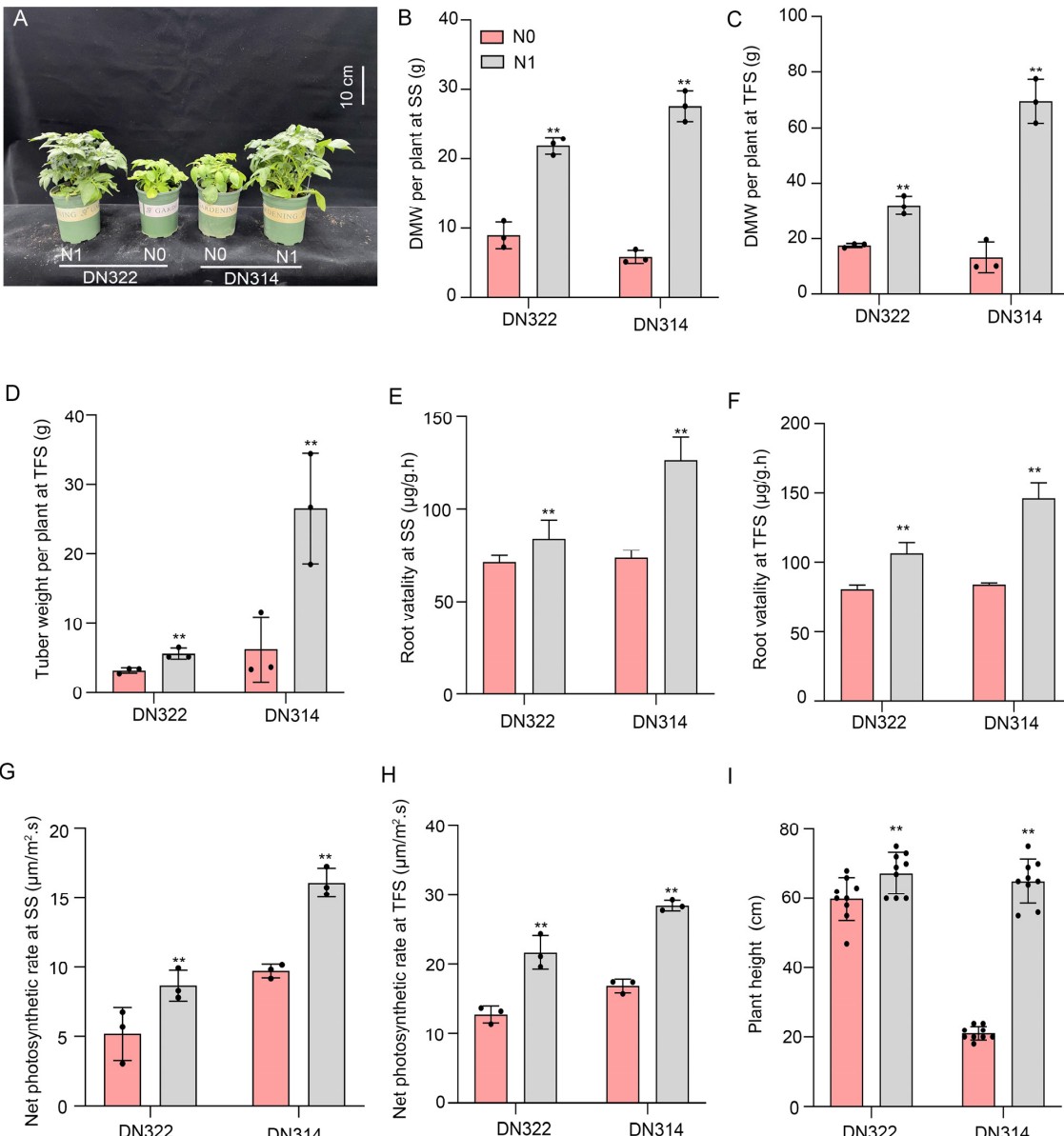

**Figure 1.** Nitrogen-deficiency stress affects potato growth and development. (**A**) Comparison of DN322 and DN314 under low- and normal-nitrogen fertilizer conditions. (**B**,**C**) Effects of nitrogen-deficiency stress on the dry matter of DN322 and DN314 at the seedling stage and tuber formation stage, respectively. (**D**) Effects of nitrogen-deficiency stress on the tuber dry matter of DN322 and DN314 at the tuber formation stage. (**E**,**F**) Effects of nitrogen-deficiency stress on the root vitality of DN322 and DN314 at the seedling stage and tuber formation stage, respectively. (**G**,**H**) Effects of nitrogen-deficiency stress on the net photosynthetic rates of DN322 and DN314 at the seedling stage and tuber formation stage, respectively. (**I**) Effects of nitrogen-deficiency stress on the plant height of DN322 and DN314. SS represents the seedling stage, TFS represents the tuber formation stage, and DMW represents the dry matter weight. Data in Figure 1 are means $\pm$ SDs. Each black dot in the figure represents the data measured using a single potato plant. A two-tailed unpaired *t*-test with Welch's correction was used for statistical analysis. Statistically significant differences ($p < 0.01$) are indicated with two stars.

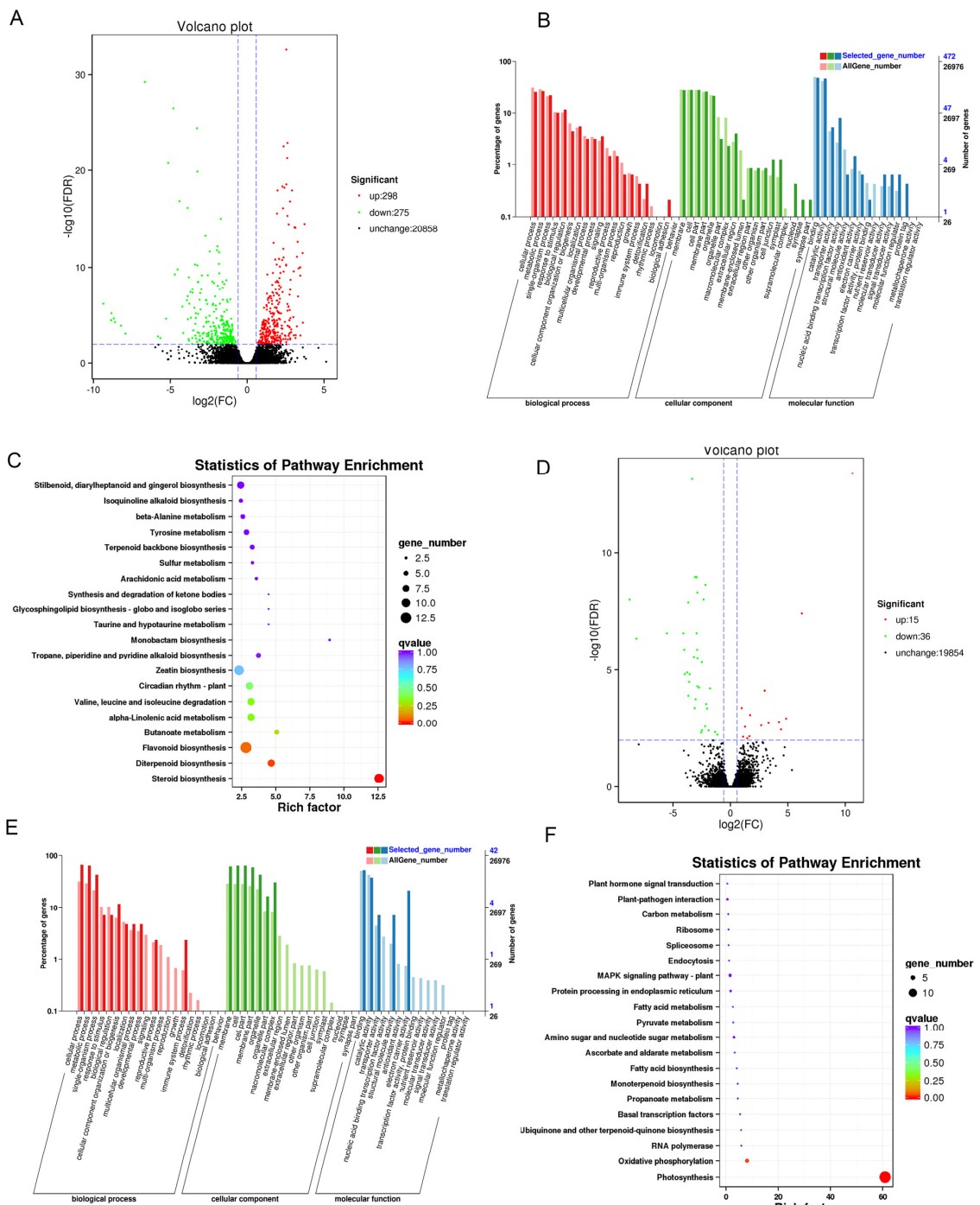

**Figure 2.** Transcriptome analysis of DN322 at the seedling and tuber formation stages under nitrogen-deficiency stress. (**A**) Volcano map of the DEGs upregulated and downregulated between N0 and N1 of DN322 at the seedling stage. (**B**) GO annotation of the total DEGs of DN322 at the seedling stage. (**C**) KEGG enrichment analysis of the total DEGs of DN322 at the seedling stage. (**D**) Volcano map of the DEGs upregulated and downregulated between N0 and N1 of DN322 at the tuber formation stage. (**E**) GO annotation of the total DEGs of DN322 at the tuber formation stage. (**F**) KEGG enrichment analysis of the total DEGs of DN322 at the tuber formation stage. Rich factor indicates the ratio of the proportion of DEGs in a pathway to the proportion of genes in all genes annotated to that pathway. The green dots represent the downregulated expression genes, and the red dots represent the upregulated expression genes in A; the larger the enrichment factor, the darker the color, and the more confident the pathway results.

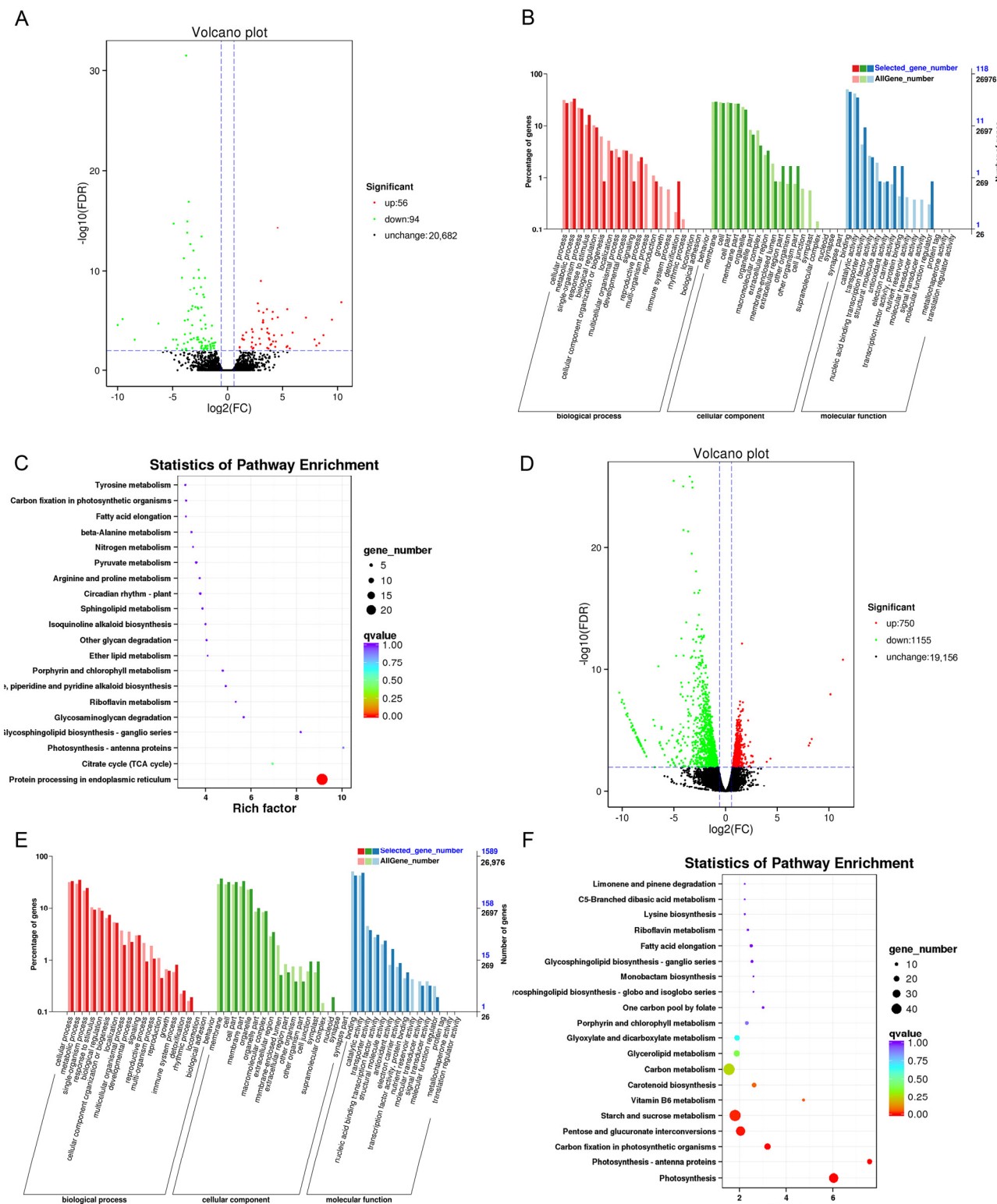

**Figure 3.** Transcriptome analysis of DN314 at the seedling and tuber formation stages under nitrogen-deficiency stress. (**A**) Volcano map of DEGs upregulated and downregulated between N0 and N1 of DN314 at the seedling stage. (**B**) GO annotation of the total DEGs of DN314 at the seedling stage. (**C**) KEGG enrichment analysis of the total DEGs of DN314 at the seedling stage. (**D**) Volcano map of the DEGs upregulated and downregulated between N0 and N1 of DN314 at the tuber formation stage. (**E**) GO annotation of the total DEGs of DN314 at the tuber formation stage. (**F**) KEGG enrichment analysis of the total DEGs of DN314 at the tuber formation stage.

*3.4. Identification of Co-expressed Gene Clusters Using Two Potato Cultivars at the Seedling Stage*

We used 12 leaf samples from the seedling stage of DN322 and DN314 for RNA sequencing. Using WGCNA, the expressed genes were divided into 14 modules (Figure 4A, Supplementary Table S8); among the modules, SM9 and SM10 were found to have the highest correlation (Supplementary Table S9), and the correlation analysis of each expression module gene's plant dry weight, plant nitrogen content, and net photosynthetic rate was carried out. SM9 and SM10 were found to have a large correlation with each phenotype (Figure 4A). The proportion of DEGs in each module was statistically counted, and it was found that the proportion of nitrogen-fertilizer-response genes in SM9 and SM10 was the largest (Figure 4B); in summary, SM9 and SM10 had the greatest relationship with nitrogen-deficiency stress. Then, we functionally annotated the genes of SM9 and SM10 modules, and through the GO database annotation, we found that the genes in module SM9 were mainly involved in the metabolic process, cellular process, localization, developmental process, and other life processes, and that these genes mainly play a role in the membrane and organelle. They mainly possess binding and catalytic activity, as well as transporter activity associations (Figure 4C). Using the KEGG database, the gene function of module SM9 was annotated, and it was found that the genes of this module were mainly annotated to the signaling pathways of plant hormone signaling and starch and sucrose metabolism (Figure 4D). Similarly, we also annotated the module SM10 genes, and through GO database annotation, we found that the SM10 genes were mainly involved in the metabolic process, biological regulation, signaling, etc., and that these genes were mainly involved in life activities in the membrane and the extracellular region, where their main molecular functions were binding, catalytic activity, transporter activity, and enzyme regulator activity (Figure 4E). Module SM10 was analyzed using the KEGG database and was found to be mainly involved in plant hormone signal transduction and protein processing in the endoplasmic reticulum process (Figure 4F).

*3.5. Identification of Co-Expressed Gene Clusters Using Two Potato Cultivars at the Tuber Formation Stage*

We used 12 leaf RNAs in the tuber formation stage of DN322 and DN314 for RNA sequencing; using the method of WGCNA, the expression genes were divided into 14 modules (Figure 5A,B), and each module was associated with the plant dry weight, plant nitrogen content, and net photosynthetic rate in the tuber formation stage. It was found that module TM1 was the most associated with these phenotypic traits (Figure 5C). We then analyzed the differentially expressed genes in each model and found that module TM1 had the highest proportion of differentially expressed genes (Figure 5D). Then, we performed gene annotation on the genes of module TM1, and via GO database annotation analysis we found that the genes of module TM1 were mainly involved in the metabolic process, cellular process, and other pathways; these genes mainly play a role in the membrane and macromolecular complex and mainly carry out binding, catalytic activity, and transporter activity (Figure 5E). These genes of the TM1 module were analyzed using the KEGG database, and it was found that they were mainly located in the photosynthetic pathway and carbon metabolism pathway (Figure 5F).

*3.6. Identification of Key Genes in the Nitrogen-Deficient Response*

In order to further identify candidate genes with regard to the nitrogen-deficient response, this study used the method of a joint analysis of DEGs and WGCNA. In the seedling stage, we used DN322 and DN314 to obtain a total of nine co-expressed differential genes (Supplementary Figure S2), and then screened the co-expressed differential genes in the nitrogen-response modules SM9 and SM10. We found that one of the nine DEGs was present in both the SM9 and SM10 modules. This gene was Soltu.DM.05G003170, a transcription factor with the zinc-finger domain, which is involved in circadian rhythm. During the tuber formation stage, we used the same method; in module TM1, a total of two genes were found, Soltu.DM.04G009820 and Soltu.DM.04G016890. The molecular

functions of Soltu.DM.04G009820 include post-translational modification, protein turnover, and chaperones, whereas Soltu.DM.04G016890 is involved in photosynthesis.

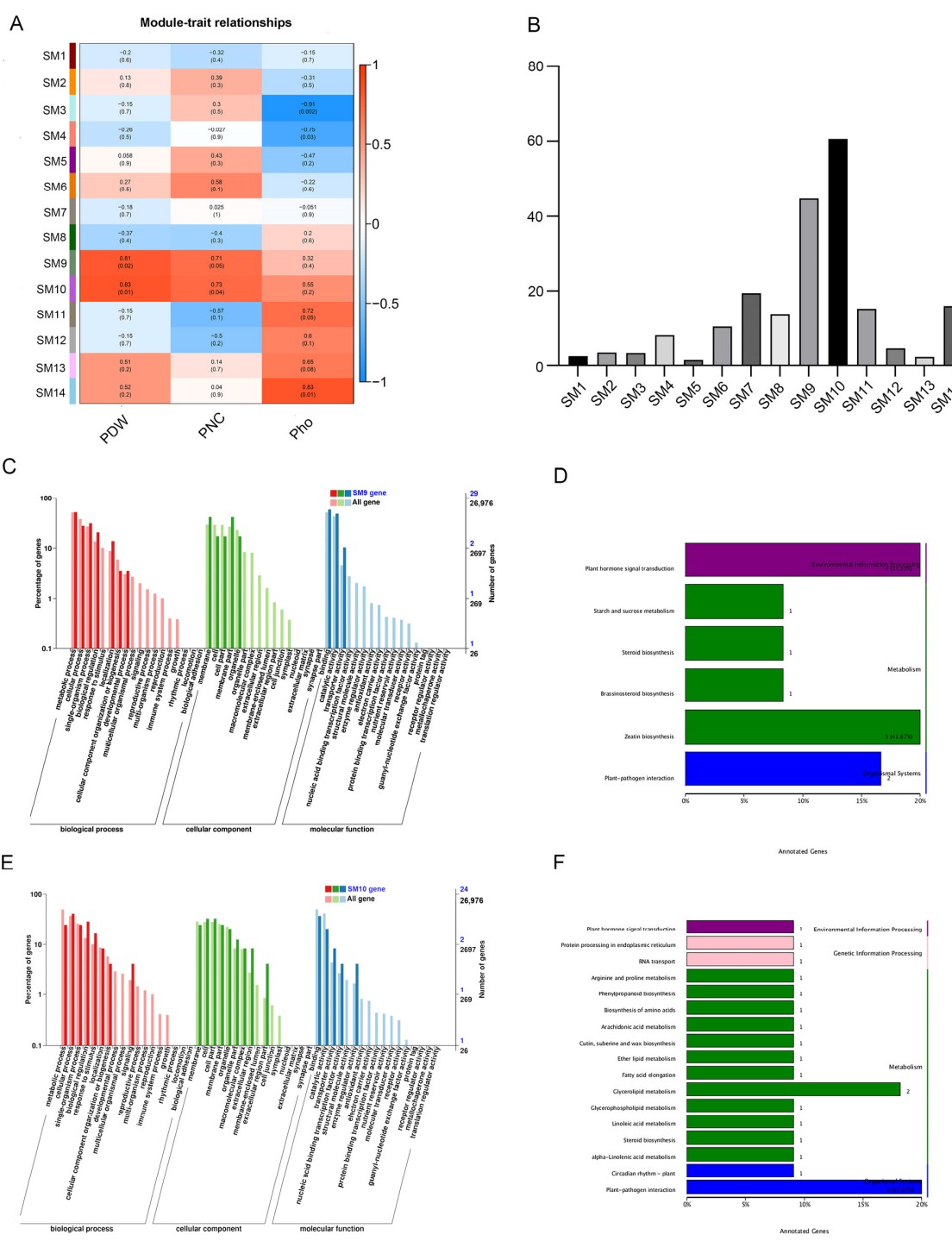

**Figure 4.** Co-expression network of two potato cultivars at the seedling stage constructed using WGCNA. (**A**) Correlation analysis of each module at the seedling stage. (**A**) Correlation analysis of each module to each phenotype at the seedling stage. PDW represents plant dry weight, PNC represents plant nitrogen content, and Pho represents the net photosynthetic rate. Red represents a positive correlation, and blue represents a negative correlation. (**B**) Proportion of DEGs in each module at the seedling stage. (**C**) GO annotation of the genes in module SM9. (**D**) KEGG annotation of the genes in module SM9. (**E**) GO annotation of the genes in module SM10. (**F**) KEGG annotation of the genes in module SM10.

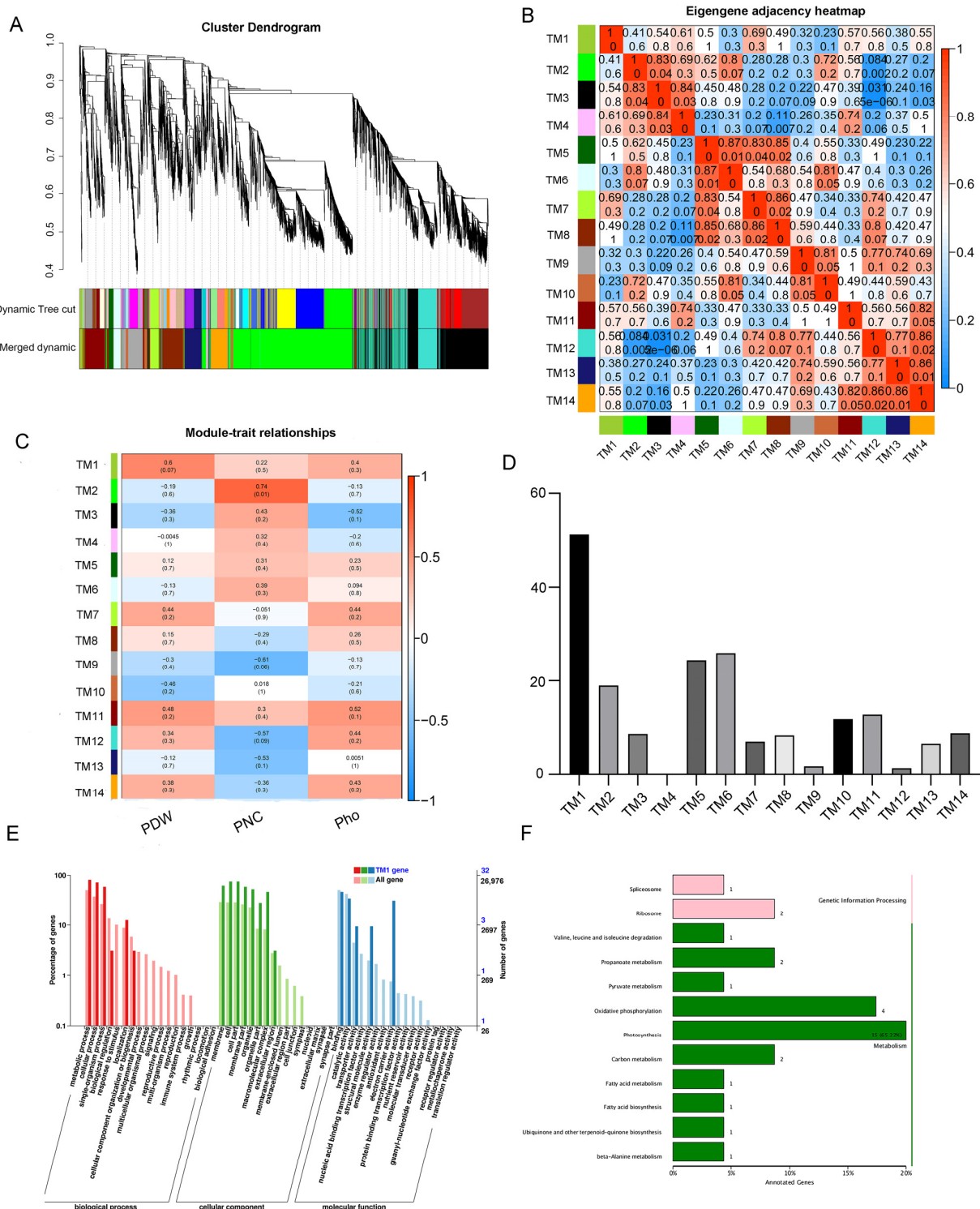

**Figure 5.** Co-expression network of two potato cultivars under the tuber formation stage constructed using WGCNA. (**A**) Overview of the co-expression network of two potato cultivars in the tuber formation stage. (**B**) Correlation analysis of each module in the tuber formation stage. The redder the color between modules, the higher the correlation between modules. (**C**) Correlation analysis of each module to each phenotype in the tuber formation stage. PDW represents plant dry weight, PNC represents plant nitrogen content, and Pho represents the net photosynthetic rate. Red represents a positive correlation, and blue represents a negative correlation. (**D**) Proportion of differentially expressed genes in each module in the tuber formation stage. (**E**) GO annotation of the genes in module TM1. (**F**) KEGG annotation of the genes in module TM1.

## 4. Discussion

### 4.1. Nitrogen-Deficiency Stress Affects Potato Growth

Nitrogen is a component of various cellular components and enzymes and plays an important role in the growth and development of plants. Nitrogen affects plant roots [36–38] and leaves [39], according to previous studies, and crop photosynthesis is also affected by nitrogen [30–42]. In this study, we first investigated the dry matter accumulation of two potato cultivars, DN322 and DN314, at the seedling and tuber formation stages under normal-nitrogen fertilizer (N1) and low-nitrogen conditions (N0), and we found that under low-nitrogen conditions, the dry matter weight of the roots, stems, leaves, and tubers of these two potato cultivars decreased significantly (Figure 1, Supplementary Figure S1, and Supplementary Table S1), indicating that nitrogen has an important effect on potato dry matter accumulation, which are the same findings as those of our previous study. We also investigated the nitrogen accumulation of DN322 and DN314 at the seedling and tuber formation stages, and found that nitrogen accumulation in the root, stem, leaf, and tuber was significantly lower under low-nitrogen conditions than that under normal-nitrogen conditions (Supplementary Table S1). Net photosynthesis and root vitality were significantly lower under low-nitrogen conditions (Figure 1E–H). In addition, DN314 was more sensitive to nitrogen fertilizer than DN322 in terms of the decrease in dry matter weight and root activity (Supplementary Figure S1). Under nitrogen-deficiency stress, the dry matter weight, nitrogen uptake, photosynthesis, and root activity of DN314 decreased more than that of DN322, indicating that DN314 was more sensitive to nitrogen fertilizer.

### 4.2. RNA-seq Is an Effective Method for the Identification of Potato Nitrogen-Deficiency-Response Genes

For seed plants, such as rice, maize, and soybeans, genes can be mapped using the QTL method by either constructing a genetic population [43–46], or using GWAS analysis methods on the natural population [47,48]. However, as potatoes are highly heterozygous and have self-incompatibility [27,49], cloning genes using QTL and GWAS methods is relatively difficult. With the development of sequencing technology, RNA sequencing technology has become an efficient method to identify candidate potato genes. In previous studies, researchers used low-nitrogen vs. normal-nitrogen or high-nitrogen vs. normal-nitrogen treatments and took samples at a certain period for RNA sequencing to obtain the DEGs, and then annotated these DEGs to determine gene function and to analyze the nitrogen-fertilizer-response genes of potatoes [50,51]. In this study, we selected two potato cultivars and selected the seedling and tuber formation stages to observe the response mechanism of potatoes to low nitrogen before and after potato tuber formation. We found that DN322 and DN314 had a total of 150 and 573 DEGs in the seedling stage, and a total of 1905 and 51 DEGs in the tuber formation stage, respectively. These DEGs can guide the study of potato nitrogen-fertilizer-response genes.

### 4.3. Effects of Nitrogen-Deficiency Stress on the Molecular Functions of Different Nitrogen-Sensitive Cultivars

Previous studies have demonstrated how nitrogen deficiency affects many pathways, such as nitrogen metabolism, transcription factors, and hormone signaling [52–54]. In this study, DN314 was more sensitive to nitrogen fertilizers than DN322. To explore its molecular mechanisms, RNA sequencing was utilized. At the seedling stage, the two varieties of enrichment pathways observed were not identical, as DN322 is mainly enriched via several biosynthetic pathways, such as steroid biosynthesis, while DN314 is mainly enriched via protein processing in the endoplasmic reticulum and photosynthesis-antenna proteins (Figure 2C and Figure 3C, respectively). Photosynthesis [55–57] is of great significance for plant growth and development, and the DEGs of DN314 are mainly enriched via the photosynthesis-antenna protein pathway, which may be an important reason underlying why DN314 is more sensitive to nitrogen fertilizer. During the tuber formation stage, the pathways with the highest degree of enrichment of DN322 and DN314 DEGs were photosynthetic pathways, while the differential genes of DN314, which are sensitive to nitrogen

fertilizer, were also enriched via the photosynthesis-antenna protein pathway and the starch and sucrose metabolism pathways. In summary, nitrogen-deficiency stress mainly affects the photosynthetic and synthesis pathways, while DN314 is more sensitive to nitrogen fertilizer as more DEGs were enriched via the photosynthetic and sucrose metabolism pathways.

### 4.4. WGCNA and DEGs Were Combined to Obtain Candidate Genes for the Potato Nitrogen-Deficiency Response

In a previous study, identifying DEGs with different nitrogen treatments provided a lot of useful information, but there were too many DEGs to make an efficient selection and these DEGs were not able to be combined with phenotypic traits [50,58]. WGCNA is an effective means for identifying co-expressed genes and for analyzing the relationship between the modules and phenotypes to obtain the key candidate genes. In this study, we performed WGCNA on genes expressing FPKM > 1 at the seedling and tuber formation stages, divided them into different modules, and correlated the modules with phenotypes. A total of 14 expression modules were obtained at both the seedling and tuber formation stages, and, interestingly, a higher proportion of DEGs were also present among the modules with a higher phenotypic correlation, indicating that these modules were associated with a low-nitrogen response. We designated these modules as low-nitrogen-response modules. Then, GO and KEGG analyses were conducted, and we found that plant hormone signal transduction and photosynthesis were enriched, as was also found in a previous study [2]. We also combined WGCNA and DEGs to obtain candidate genes with a low-nitrogen response. There were nine and four co-DEGs between DN322 and DN314 (Figure 3E) at the seedling and tuber formation stage, respectively. Then, co-DEGs and low-nitrogen-response modules were combined to obtain one and three genes associated with low-nitrogen stress; these genes will be the focus of future research.

### 5. Conclusions

At the seedling stage, low-nitrogen stress mainly affects potato growth by affecting the photosynthetic pathway and the steroid biosynthesis pathway. The tuber formation period mainly affects the photosynthetic pathway and the starch and sucrose metabolism pathways. In this study, four genes associated with low-nitrogen stress were identified through WGCNA combined with the study of differentially expressed genes (Figure 6), which can provide theoretical guidance for the high nitrogen efficiency breeding of potatoes.

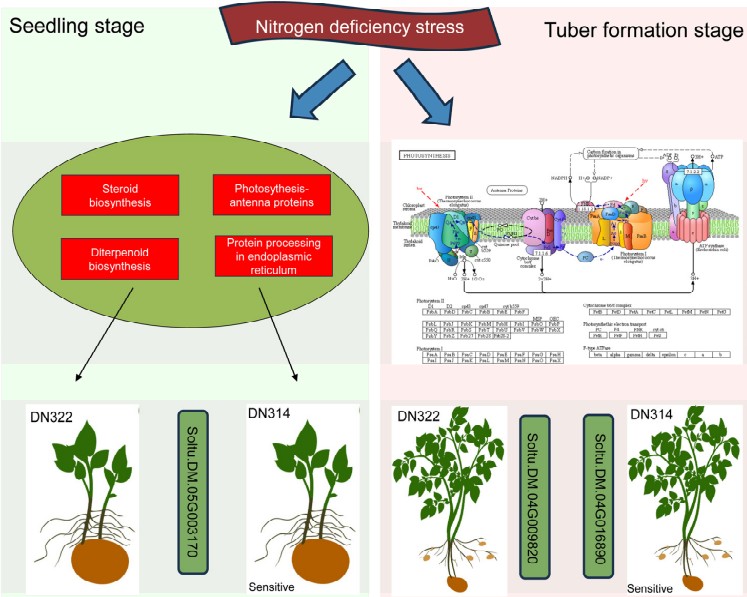

**Figure 6.** The model employed in this study. On the left is the seedling stage: When nitrogen-deficiency stress occurs, the DEGs of DN322 are mainly enriched via the steroid biosynthesis pathway and diterpenoid

biosynthesis, whereas DN314 DEGs are mainly enriched via the photosynthesis-antenna proteins and protein processing in the endoplasmic reticulum pathway. The gene Soltu.DM.05G003170 was identified in response to nitrogen-deficiency stress through combined analysis of WGCNA and DEGs. On the right is the tuber formation stage: When nitrogen-deficiency stress occurs, the DEGs of DN322 and DN314 are mainly enriched via the photosynthesis pathway. Two genes, Soltu.DM.04G009820 and Soltu.DM.04G016890, were identified in response to nitrogen-deficiency stress through combined analysis of WGCNA and DEGs.

**Supplementary Materials:** The following supporting information can be downloaded at https://www.mdpi.com/article/10.3390/agronomy13082164/s1. Figure S1. Effects of nitrogen-deficiency stress on the potato. Figure S2. The RT-qPCR verification of DN322 and DN314 and DEGs analysis. Table S1. Nitrogen-deficiency stress phenotype. Table S2. RNA-seq quality of DN322. Table S3. RNA-seq quality of DN314. Table S4. The DEGs of DN322 between N0 and N1 at the seedling stage. Table S5. The DEGs of DN322 between N0 and N1 at the tuber formation stage. Table S6. The DEGs of DN314 between N0 and N1 at the seedling stage. Table S7. The DEGs of DN314 between N0 and N1 at the tuber formation stage. Table S8. Proportion of DEGs in each WGCNA module at the seedling stage. Table S9. Proportion of DEGs in each WGCNA module at the tuber formation stage.

**Author Contributions:** Z.G. and Q.W. designed this study; Q.W. and B.D. determined the dry matter weight and physiological indices; B.D. and X.S. took the sequencing samples; Y.Y. analyzed the data and wrote the paper; and Y.S. provided the potato materials. All authors have read and agreed to the published version of the manuscript.

**Funding:** This work was supported by the earmarked fund for the China Agriculture Research System (CARS-09).

**Data Availability Statement:** All data related to this manuscript can be found within this paper and its Supplementary data.

**Conflicts of Interest:** The authors declare that the research was conducted in the absence of any commercial or financial relationships that could be construed as potential conflict of interest.

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
