# Peer review of "Transcriptome Analysis of Nitrogen-Deficiency-Responsive Genes in Two Potato Cultivars"

_agronomy, doi:10.3390/agronomy13082164_

Round 1

Reviewer 1 Report (Previous Reviewer 2)

Dear Authors,

Thank you for editing and response. I can tell your efforts in this new version. 

I would suggest accepting in this version. 

Thank you. Good luck.

Author Response

Thank you for your kind comment and good luck.

Reviewer 2 Report (New Reviewer)

Wei et al conducted transcriptome analysis of nitrogen-deficiency responsive genes in two potato cultivars. The manuscript provides information. However, there are few crucial points that if considered will increase the accuracy, value of the manuscript and may be readability.

-First of all, please add the novelty of the work, if there is any to the abstract and introduction

-English language is an issue in the entire manuscript. For example: Line 15; Line 19: replace applied with utilized. Line 26 etc. Please thoroughly correct the entire manuscript for such mistakes.

-Please add the future prospect of the study to the abstract.

-Which growth media was present in pots, soil, sand or clay?

-From Line 83 to 89, please briefly explain all the ‘previous methods’.

-There is no information about the RNA sequencing in the methodology. You can follow and cite other studies such as 10.3390/agronomy13030631 to add relevant information.

- How many replicates were used for RNA isolation and RT-qPCR?

-There is no information on quality and quantity of isolated RNA. For example see the manuscript (10.3390/agronomy12102421) to add such information.

-In RT-PCR methodology, add how many genes were studied?

-There is no statistical information on the conducted analysis.

-Was there any difference in the NUE of the two experimental genotypes, DN322 and DN314?

-Why leaves were chosen for RNA sequencing? Please add.

-I could not find the RT-PCR results.

-Please improve the discussion by adding references from the previous studies.

-Tables and Figure captions should be self-explanatory. Please elaborate them by mentioning the experimental plant name, treatments etc. Applicable for all the captions.

I do believe that the manuscript can be accepted once the authors address the mentioned points and enrich the manuscript with the crucial information. 

-English language is an issue in the entire manuscript. For example: Line 15; Line 19: replace applied with utilized. Line 26 etc. Please thoroughly correct the entire manuscript for such mistakes.

Author Response

This manuscript is a resubmission of an earlier submission. The following is a list of the peer review reports and author responses from that submission.

Round 1

Reviewer 1 Report

- M&M section:

How many reps, what was considered an experimental unit? How was this considered in the stats analysis?
What was the volume of the pots? and the subsequent N concentration in the soil, what type of substrate was used?

RNAseq sampling: Why only after 12 and 24 d ? why not later? And why not in the tubers? That is the most relevant tissue.

Throughout MS: use space before ,( and other signs.

Many sentences are hard to interpret: e.g. line 170-171

Probably fig 2 anmd 3 cna be combined into one figure of both cultivars. the figure is hard to read, clearer axes are needed.

Line 285-287 is not true, many GWAS/QTL studies in potato are available

Sentences miss a good structure, e.g line 170-171; Lines 110-111, also section lines 75-81 needs revision of english.
Use of verbs should be checked: e.g. line 83; line 96

Use a space before each special character: (;, etc.

Reviewer 2 Report

Dear Author,

Thank you for your work. Authors probably prepared this manuscript very rush. There are many minor details that haven’t been corrected before submission.

Overall, author needs to improve the expression in English. Maybe you will find a professional team to correct grammar, punctuation, and reference format, such as Line#27-#29, Line#38 (reference 10), Line#50 et al.

Line#3: Please check on the name list whether need to add a name into or delete “and”.

Line#16: Checking the space of “RNA -seq”. RNA-seq would be a correct version.

Is weighted correlation network analysis (WGCAN) in Line#54 the same as WGCNA in Line 21?

Can you explain the Rich factor in figure 2C and F?

Maybe authors provide high-resolution figures. 

I would suggest improving the English expression. 

Reviewer 3 Report

Totally, I suggest the authors should rewrite the abstract and improve its coherence. The aim of the study is not clear. In addition, a more informative introduction needs to make your aim of study stronger. The materials and methods are not clear and need to be written with more details. The results lack a proper presentation and discussion. Moreover, I suggest authors should revise and improve English language. Finally, where is the conclusion of the study??????

Sorry that I am not able to be more positive about this manuscript.

I suggest authors should revise and improve English language.

Round 2

Reviewer 1 Report

Important points of commentary have been corrected.

adequate